# Real-Time Sensor-Based Human Activity Recognition for eFitness and eHealth Platforms

**DOI:** 10.3390/s24123891

**Published:** 2024-06-15

**Authors:** Łukasz Czekaj, Mateusz Kowalewski, Jakub Domaszewicz, Robert Kitłowski, Mariusz Szwoch, Włodzisław Duch

**Affiliations:** 1Aidmed, 80-254 Gdańsk, Poland; mat.kowalewski98@gmail.com (M.K.); jdomaszewicz@aidmed.ai (J.D.); robert@aidmed.ai (R.K.); 2Faculty of Electronics, Telecommunications and Informatics, Gdańsk University of Technology, 80-233 Gdańsk, Poland; szwoch@eti.pg.edu.pl; 3Department of Informatics, Institute of Engineering and Technology, Faculty of Physics, Astronomy & Informatics, Nicolaus Copernicus University, 87-100 Torun, Poland; wduch@is.umk.pl

**Keywords:** human activity recognition, human–computer interaction, deep networks, contrastive learning, mobile, inertial measurement unit

## Abstract

Human Activity Recognition (HAR) plays an important role in the automation of various tasks related to activity tracking in such areas as healthcare and eldercare (telerehabilitation, telemonitoring), security, ergonomics, entertainment (fitness, sports promotion, human–computer interaction, video games), and intelligent environments. This paper tackles the problem of real-time recognition and repetition counting of 12 types of exercises performed during athletic workouts. Our approach is based on the deep neural network model fed by the signal from a 9-axis motion sensor (IMU) placed on the chest. The model can be run on mobile platforms (iOS, Android). We discuss design requirements for the system and their impact on data collection protocols. We present architecture based on an encoder pretrained with contrastive learning. Compared to end-to-end training, the presented approach significantly improves the developed model’s quality in terms of accuracy (F1 score, MAPE) and robustness (false-positive rate) during background activity. We make the AIDLAB-HAR dataset publicly available to encourage further research.

## 1. Introduction

Human Activity Recognition (HAR) is focused on the recognition of specific human movements and actions using data from various sensors and usually involves challenging time-series classification tasks. HAR may be performed at different levels of granularity: from general types of activity (*actigraphy*) like walking, sitting, sleeping, standing, showering, or cooking, to a more fine-grained detection of particular exercises (such as push-ups or squats), repetition counting, and motion-based game controls and natural user interfaces. Various approaches to HAR may be classified into two categories, vision-based or sensor-based, with an Inertial Measurement Unit (IMU) as one of the popular sources of signals. The computational approach to time-series classification in the case of sensor-based HAR may be done using hand-crafted features, Dynamic Time Warping (DTW), or deep neural models. Supervised machine-learning methods, such as logistic regression, support vector machine, k-nearest neighbors, and other methods, have been used to study activity recognition based on popular wrist-worn IMUs [1]. Recently, standard Convolutional Neural Networks (CNNs) have also been used to count repetitions for many types of exercise, with great success [1,2]. A comprehensive review of these approaches has been presented in [3,4,5,6].

HAR methods enable detailed insight into specific movement patterns and their automated quantification, which is valuable, especially in the case of remote monitoring and long-running rehabilitation scenarios. Therefore, an efficient HAR is the basis for the development of remote assistance tools with applications in exercise-based telerehabilitation [1,7,8], remote monitoring and remote examination of patients [9,10], sport and fitness [2,11,12,13], or gaming [14,15,16]. The great interest in such technologies results from the fact that they provide an objective way for tracking participant/patient adherence to prescribed training/rehabilitation plans, measuring the volume of their activity and progress. This allows for optimization of the training/rehabilitation plans and faster recovery [17,18]. From the participant/patient perspective, such technologies are the source of valuable continuous feedback in the absence of direct contact with physiotherapists or trainers. Combined with gamification techniques, this helps to build motivation for adherence to the prescribed training programs and setting the exercise pace.

This paper presents some results from the project aimed at building an easily-extendable HAR software framework (AIDLAB-HAR 1.0). The following use cases are planned for our framework:mobile application for remote/at-home observation of elderly people or people with chronic diseases (e.g., COPD, long-COVID);remote/at-home testing (Fullerton Test [19], sit-to-stand test [10], remote stress tests [20]);telerehabilitation (physical, pulmonary, cardiac): feedback and adherence monitoring;promoting physical activity for elderly people;gamification in telerehabilitation and activity promotion for young people (smart games).

Note that, due to safety concerns, these use cases require the integration of HAR with cardiac monitoring/pulse-oximeter. These use cases impose the following requirements on the framework:HAR should be based on wearable IMU sensors (ideally one);it should ignore background activities (low rate of false-positive detection);it should be easy to tune for users with different levels of physical ability;full-body activities should be recognized;it should be easy to integrate our framework with mobile applications (iOS, Android) and provide real-time feedback;it should be easy to add new exercise/movement patterns with a small number of examples;end users are not expected to be IT specialists, so no manual feature engineering should be required if a new exercise is added.

In our work, all signals were recorded with a unique portable Aidmed One recorder [21]. We plan to use this framework in the commercial Aidmed telemedical system (aidmed.ai). However, our approach can be extended to other wireless IMU devices (e.g., MbientLab, Shimmer, Polar H10).

In this paper, we focus on counting repetitions of selected types of exercises. We assume that the type of exercise is known in advance due to the training plan or game scenario. Therefore, the problem can be treated as a recognition of exercise patterns in the time series [22]. We apply post-processing of the output (score) provided by the detector (i.e., score threshold, refraction time) to better fit the loss measure, equal to the difference between the performed and detected number of repetitions. Further details are described in Section 2.

Several papers have used deep-learning techniques in application to HAR (for a summary, see Table 1). In [2], signals recorded simultaneously from two smartwatches are used for the recognition and repetition counting of 10 complex full-body exercises typical in CrossFit (e.g., pull-ups, push-ups, burpees). For recognition, the authors apply a deep neural network consisting of convolution layers followed by dense layers. Overlapping windows of 5 s of raw sensor data are classified by the recognition network. Then, the onset of repetition is detected with an exercise-specific deep network. This network is selected on the basis of results from the recognition network, and it has an architecture similar to the network in the recognition step. Repetition counting is performed on the sequence of labels provided by the detector at the beginning of repetition. Recognition and repetition counting are performed offline. During data collection, repetitions of the exercise were performed on demand, and vibrations of the smartwatch signaled the start of the repetition. After exhaustive hyper-parameter optimization, this method achieved a classification accuracy of 99.96% and repetition counting within an error of ±1 repetitions in 91% of the tests.

A similar problem is discussed in [1]. Here, the authors describe a method for the recognition and repetition counting of 10 endurance-based exercises (e.g., biceps curls, squats, lunges) on the basis of signals from a single wrist-worn IMU sensor. The exercise recognition task was treated as a multi-class classification task with a deep CNN approach based on AlexNet architecture. The repetition counting is based on the results of the classification task and counts compact segments of detection. The deep CNN approach was compared with classical machine-learning methods, such as support vector machines, random forest, k-nearest neighbor, and multilayer perceptrons. Deep CNN networks and classical methods are fed with signals from the 4 s sliding window. Peek detection for repetition counting is performed offline on the whole series. Researchers compared repetition counting from the dominant accelerometer axis and exercise detector output. The reported quality of the deep CNN approach was high; the F1-score was equal to 97.18% for exercise recognition and ±1 repetition error among 90% observed series for repetition counting.

Besides the deep-learning approaches, the following works, in which the authors rely on classic machine-learning methods, are worth mentioning. In [23], random forest was used for recognition between five activities (regular walking, climbing stairs, talking with a person, staying standing, working at a computer). The classifier used 20 features efficiently computed in 1 s windows of signals from a sensor located on the chest. The paper reports 94% accuracy of human activity recognition. The authors do not study repetition counting.

The phone-based body sensor network myHealthAssistant [24] classifies gym exercises from three accelerometers (on the hand, arm, and leg), using a Bayesian classifier trained on the mean and variance on each accelerometer axis. The paper reports 92% accuracy for 13 exercises in the subject-specific training scenario. Repetition counting is based on peak counting on one of the accelerometer axes.

RecoFit [11] presents a pipeline of three tasks: segmenting exercise from intermittent non-exercise/rest periods, exercise recognition, and repetition counting. For segmentation and recognition tasks, the authors use linear support vector machine computing features from 5 s segments of signals from the sensor on the arm. Repetition counting is performed offline using peak counting on a synthetic signal obtained from an acceleration vector. Precision and recall were greater than 95% in identifying exercise periods, accuracy was between 96 and 99% for exercise recognition (depending on the series length), and counting was accurate to ±1 repetition in 93% of series from the dataset of 26 exercises.

Our approach differs from the earlier works [1,2,11,23,24] in several important aspects:we use a single sensor placed on the chest, while in the cited papers the sensor is placed on the wrist, or multiple sensors are used;our approach works in real-time, while repetition counting in the cited papers is performed offline, has high latency, and relies on peak detection for the whole series; to our best knowledge, ours is the first evaluation of real-time repetition counting with a chest-placed IMU sensor; on the other hand, real-time operation mode does not reach as high quality as [2,11];we use one deep network model with encoder–detector architecture for all types of exercises; compared to [2], our solution is easier to extend to new exercises and is suitable for mobile devices.

Novel contributions of our work include:a deep neural network model for real-time exercise recognition and repetition counting based on signals from a chest-located IMU sensor;a method of false-positive error reduction based on contrastive learning;publicly available dataset AIDLAB-HAR to encourage further research on this topic.

For the purpose of the project, we developed a mobile application that supports data acquisition, guiding users through the workout plan. We used our in-house utility software to visualize and annotate collected signals. We have also developed a weak supervision algorithm to speed up and simplify the annotation process.

## 2. Materials and Methods

### 2.1. Data Collection

Data were collected from volunteers during CrossFit or functional training. The workout consisted of a series of exercises (see Section 3.1). Each series consisted of a fixed number of repetitions (e.g., 10) of a given exercise. Some series were also limited by a fixed duration time. Series took 30–90 s and were separated by 30–60 s of rest. Each series was performed 3 times. Exercises were interlaced, i.e., there was no consecutive series of the same exercise. The workouts were performed with the support of a professional instructor. Before data collection, the instructor demonstrated how to perform the exercises and use the recorder and mobile app, and introduced the workout plan. Participants were told to exercise at their own pace. The signals were collected from the whole series of exercise repetitions performed without breaking the sequence or doing exercises on command. Such an approach to data collection better reflects real-world HAR usage than a series of single on-demand exercises.

Data collection to assess model quality was designed to handle the following cases:differentiate between similar exercises (e.g., crunches vs. abdominal tenses, lunges vs. side lunges);full body exercises (e.g., burpees, standing-to-plank-downward-dog-to-plank sequence), exercises used in tests (sit-to-stand).

Signals were collected using the Aidmed One recorder (see Figure 1 and [21] for details). Data from the recorder were received via Bluetooth by a mobile application and then transmitted to a server. Besides data retransmission, a mobile application was a guiding assistant, instructing our participants about upcoming workout routines or rest periods after each series. Participants marked the start and end of the series/rest, thus segmenting recorded signals into a series of exercises. Participants were verbally informed of the number of repetitions/duration/end of each activity. The mobile application allowed the marking of single repetitions. However, this function was only used when the instructor assisted the participant, observing and marking the end of each particular exercise repetition without disturbing the participant. All marks were synchronized with signals and sent to the server. Data collected from the sensor included acceleration in the recorder frame, quaternion of orientation (sensor fusion from the 9-axis was done on the IMU), ECG, and respiration signals. However, the results reported below are based only on the acceleration and orientation signals.

Annotations used in the training and evaluation were obtained in the following way. First, reference segmentation of the series of exercises into repetitions (based on marks done by the instructor) was presented to the annotators. Next, the annotators segmented the series of signals from the exercise. Finally, annotations were obtained as events centered on fiducial points of the repetitions inside a given segmentation of the series. Fiducial points were, for example, peaks/valleys of the signal in the most informative channel, i.e., the channel with a clear visual repetition of the signal. Annotations were 0.2 s wide, with a 0.1 s margin on each side. There were no overlapping annotations in the dataset. There were at least 0.5 s of separation between annotations. See Figure 2 for details. This approach helped to standardize annotations and improve the quality of our models (cf., [25] for annotation quality issues).

To speed up and simplify future annotation processes, we implemented the following steps in the web application algorithm:annotator provides 1–3 reference annotations of repetitions of a given exercise;annotator selects informative signals (one or more);DTW [26] is calculated for each informative signal between reference annotations and window sliding on the data series of a given exercise;for each window, distance is calculated as the median of values obtained in step (iii);for a given series, threshold is calculated as a fixed fraction of the median of window distances in this series;for a given series, all distance minima below the threshold are selected as repetitions.

The algorithm was validated with two reference annotations on 6 types of exercise (abdominal tenses, crunches, squats, lying hip rises, bends, push-ups) and obtained a mean recall equal to 1.0 and mean precision equal to 0.93 (push-ups had the worst performance with precision of 0.87). We decided to maximize recall because removing a wrong annotation is easier than adding a missing one.

Signal prepossessing involved the following steps (see Figure 1a):raw signal collected from the device consists of recorder frame acceleration along the X, Y, and Z axis, and the rotation quaternion; data were collected at 50 Hz;for further processing, we take recorder frame acceleration and calculate linear acceleration along the Z axis (of the earth frame), adding pitch and yaw rotations;signals are filtered using the low pass filter with a cut-off frequency of 10 Hz;filtered signals are arranged in windows of 2.8 s size, sliding in 0.1 s steps (see Figure 2);data in each window is given as input to the deep neural network.

### 2.2. Detector

The neural network model used to analyze activity is fed with signals organized in windows of 140 consecutive samples (2.8 s) from 6 channels. Detection is performed every 0.1 s.

The neural scoring model consists of two parts (see Figure 3): encoding network and classification head. The encoding network starts with the convolution part, performed by a stack of the following layers: 1D convolution layer (kernel 5×1, 48 filters, 8/channel); 1D convolution layer (kernel 3×8, 48 filters, 8/channel); 1D convolution layer (kernel 1×48, 48 filters); max pooling layer (pool size = 5), spatial dropout = 0.1.

The results of the convolution parts feed two stacked bidirectional LSTMs (Long-Short Term Memory), each of size 32. The final states in both directions of the top LSTM are concatenated and provided as input to the classification heads. This part of the model consists of two hidden dense layers, each of size 64, and the output layer of size 1.

We reduce the detector variance by stacking 2 neural models trained with different data augmentation and different initialization of weights and averaging their outputs. Stacking more than 2 models had no significant effect on the obtained results.

We experimented with other architectures (e.g., CNN without LSTM, CNN + forward only LSTM, XGBoost [27] model with manual features), but they had a lower performance. We also experimented with the replacement of the classification head with the k-nearest neighbors algorithm. This approach also gave lower performance and introduced additional complexity in implementation on mobile devices.

The output of the neural model is post-processed to reduce false-positive detections. See Figure 1 for details.

### 2.3. Training

In this paper, we compare two methods of training.

Training the encoder with contrastive learning and Euclidean distance, fixing encoder parameters, and training the classification head with binary cross entropy loss. We train one encoder for all exercises and use a dedicated classification head for each type of exercise.End-to-end training, where the encoder and the classification head work as a single model and are trained together with binary cross entropy loss. In this case, we train a dedicated model for each exercise.

In this paper, we show that contrastive learning [28] reduces the false-positive rate and facilitates few-shot learning [29]. The signals provided to detectors are organized in data windows of 2.8 s length. Each window is described by exercise type and a ‘repetition’/‘background’ label. See Figure 2 for details.

We performed data augmentation (cf., [9]) using 3 natural transformations: rotation of the reference frame (constant for the whole window), global time scaling, and local time scaling. We added 9 augmented examples to the original one.

For contrastive learning, we treat a pair of data windows as positive if the exercises and their labels match. An equal number of positive and negative pairs is collected into each batch. For training classification of the head and end-to-end model for a given exercise, we take repetition examples of that exercise as positive, while background examples and all examples from other exercises as negative. We balance positive and negative examples with negative class weight calculated from the proportion of frequency of positive and negative examples.

Data from all exercises was used in the training detectors, classification heads, and end-to-end models. In both cases, detectors trained only on the examples from the given exercise had significantly worse quality. A high false-positive rate (especially during preparation for exercise) made them unusable in practice. Extending the dataset with plank and running plank as the background for push-ups improved the quality of the detector. We also observed the positive effect of extending the dataset with walk examples on the quality of jumps, squats, and lunges detectors.

In addition to the learning approaches described above, we tested two similar learning methods, based on triplet loss and contrastive loss with cosine distance, with no significant improvement.

## 3. Results

### 3.1. Dataset

We built a dataset of 15 activities recorded during functional training (abdominal tenses, standing-to-plank-downward-dog-to-plank sequences, lying hip rises, side lunges, sit-to-stands) and during CrossFit training (broad jumps, burpees, crunches, lunges, push-ups, squats, planks, running planks, and walk). We built detectors for the first 12 activities, while the last three activities served only as the additional source of background examples.

Data were collected from 24 participants aged 20–45: 15 participants (mainly men) performed approximately 20 repetitions of CrossFit exercises in two series, while the other 9 participants (mainly women) performed 30 repetitions of functional training exercises in three series. In some series, there were missing or redundant single repetitions.

### 3.2. Evaluation

We evaluated detectors in a 3-fold cross-validation schema. We randomly split users between the training and test sets; no user appeared in both sets simultaneously in the same fold. The approach may be treated as user-agnostic. For a given exercise, we evaluate detectors by calculating the F1 scores and the mean absolute percentage error (MAPE). These metrics were calculated according to the annotated repetitions. For F1 score, we assume that detection matches the annotation if it fails in the annotation range extended with a 0.1 s margin. We perform a matching algorithm to ensure that one detection matches no more than one annotation. For MAPE, we compare the number of detections and number of annotations in a given series of exercises (10 repetitions on average). We use macro-averaging to integrate results from different series. Each detector was also evaluated using the other exercises to assess its robustness in the background activity. We calculated the false-positive rate (FPR) for each background exercise as the mean number of detections per one second of the signal. We took the maximal FPR value among background exercises to assess worst-case robustness.

The results of the evaluation are presented in Table 2 (we present results only for the best architecture described in Section 2.2). The first column contains exercise type and training method: ‘enc’ refers to the model with a pretrained encoder and ‘e2e’ refers to the end-to-end training. In the other columns, we present the quality metrics. F1 and MAPE are expressed in percentage, and FPR is expressed in detections per second. For each measure, we provide a median value from the 3-fold and interquartile range (in brackets). There is a small variation between folds.

The results presented in Table 2 show that the encoder pretrained with contrastive learning significantly reduces FPR for background activity. For some exercises, the encoder can also increase the quality of detection (F1, MAPE). In general, models have a high detection rate and acceptable FPR. Based on our observations, we estimate that 10 s is sufficient to prepare for the exercise and mark the end of the series after it is finished. It is also the reasonable distance between two activities in activity-based games. The worst performance was obtained for standing-to-plank-downward-dog-to-plank sequences, where our models are not usable in practice. One of the reasons for this may be that this is quite a long (>6 s) multi-step exercise with low dynamics. For bends, we observed a large variance in flexibility/range of motion between participants, and for push-ups in the dynamics and range of motion. An extension of datasets with planks and running planks as background activity positively affected the quality of push-up detection.

We implemented the data pipeline and models with Python 3.8 and TensorFlow 2.6. We decided to base a deep-learning model on *TensorFlow*/*TensorFlow Lite* since this platform offers a streamlined process for building models and deploying them on mobile devices. The mobile application was written in Flutter and achieved a performance of 100 (detections/s) in our tests on a mid-priced phone.

### 3.3. AIDLAB-HAR Dataset

We provide part of the collected dataset with the intention of boosting research on HAR. The dataset consists of annotated recordings from various exercises and activities. Recordings contains recorder frame acceleration (signal labels: $acc_{X}$, $acc_{Y}$, $acc_{Z}$) and quaternion of orientation ($q_{X}$, $q_{Y}$, $q_{Z}$, $q_{W}$). All signals are recorded with a frequency 50 Hz and synchronized. Signals are stored in *EDF* format. The naming convention for the files is *SUBJECT_EXERCISE_SERIES.(edf|csv)* (e.g., SUB1_SQUAT_S1.edf), where *.edf* are files with signals and *.csv* are corresponding files with annotations. The annotation file contains two columns: *TIMESTAMP* (in seconds, from the start of recording) and *EVENT* (*SERIES_ONSET*, *SERIES_OFFSET*, *REPETITION_ONSET*, *REPETITION_OFFSET*).

There are 13 types of exercise (*ABDOMINALTENSE*, *BEND*, *BROADJUMP*, *BURPEE*, *CHARISTANDANDSIT*, *CRUNCH*, *DOWNWARDDOG*, *LUNG*, *LYINGHIPRISE*, *PUSHUP*, *SIDELUNGE*, *SQUAT*, *ROTATINGTOETOUCHE*) and 3 types of background activities (*WALK*, *PLANK*, *RUNNINGPLANK*). Each type of exercise has recordings from five subjects and two series for each subject. There are 10 repetitions for each series on average. For background activities, there are recordings from 10 subjects. For each subject, there are >300 s of *WALK* and two blocks of *PLANK*, *RUNNINGPLANK* with durations >20 s.

Subjects were healthy volunteers: all men, aged 20–40 years. Data are available under https://www.aidlab.com/datasets (accessed on 6 May 2024). The repository contains Python scripts for signal preview (*data_preview.py*). Signals and annotations are in the ‘data’ folder.

## 4. Discussion

The primary goal of our work was to create real-time detectors and repetition counters for 12 types of exercise using a single IMU sensor placed on the chest. We have shown that our system can be used in practice to obtain effective and reliable results. Deep neural models based on stacked encoders and LSTM were used to classify acceleration and orientation signals from our sensor, providing automatic annotations that can be used to evaluate the performance quality of CrossFit workouts and functional exercises. These models were implemented on mobile devices and can be used to monitor rehabilitation progress and individual training. Their accuracy has been compared to other classification methods and network architectures. We have highlighted contrastive learning as an effective quality improvement method in the HAR area.

The reason behind quality improvement with contrastive learning is that it enables the encoder to learn better representation where examples from different exercises consist of well-separated clusters. The representation is more robust for individual differences in exercise performance and focuses on more details differentiating similar exercises. We conducted qualitative experiments, trying to understand how detectors perform on exercises sharing similar parts of motion (e.g., broad jumps and squats). We have tried to cheat detectors by performing different variants of movement and position changes (e.g., from lying to standing) similar to those in the original exercise. We observed that the encoder-based detector is more robust for that kind of adversarial attack, while the end-to-end detector is more focused on the detection of fiducial points that lead to a higher false-positive rate.

In our approach for repetition detection and counting, only a 2.8 s window of the latest signal is used. There is no need for prior exercise series recognition or searching through the whole series for detection peaks. This contrasts our method to the previous works described in the Introduction [1,2,11,23,24], where a significantly longer window (>5 s) and a whole time series is used for analysis. The short data window and, as a consequence, the short response time of the detector, as well as the implementation on the mobile platform, show that the present method is suitable for real-time applications. Since our method is robust against false-positive detection when changing exercises, it may be used to detect single repetitions and find application in game control with user motion. Another advantage in comparison to [1,2,11] is the location of the sensor on the chest, which allows for recording reliable ECG signal and may be used for patient monitoring during exercises.

Collecting experimental data for HAR experiments takes a long time. The application of our models to similar experiments will not allow us to draw conclusions about the efficiency of our models, as the results may strongly depend on the type of sensors used. We have released a novel annotated dataset of 13 different exercises and three background activities. This should enable other researchers to compare the results of their methods with the best models reported in this paper.

Although only two types of signal were used to create our classification models, the other signals from Aidmed One contain an additional diagnostic value. This sensor can also measure changes in the chest volume (bioimpedance), analyze breathing patterns using a microphone (e.g., coughing), measure skin surface temperature, collect ECG signals using silicone electrodes without gel, and measure pulse rate and oxygen saturation of the blood (SpO2 sensor). It can also work with the pressure sensor to measure airflow through the nose/mouth. Such data transmitted wirelessly during physical exercises may be very useful for an overall assessment of many health conditions, enabling evaluation of the general physical condition of the trainees, including the fitness of sports trainees. Our platform will be further developed to integrate all such data into useful clinical biomarkers.

## Figures and Tables

**Figure 1 sensors-24-03891-f001:**
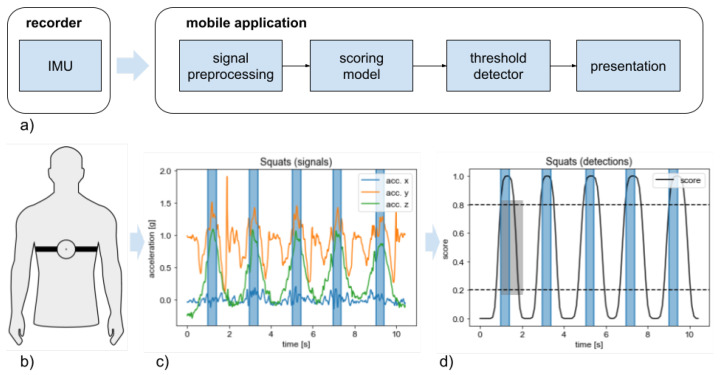
The proposed signal pipeline from the recorder to the repetition counting: (**a**) block diagram of the pipeline; (**b**) recorder location; (**c**) an example raw signal representing 5 repetitions of squats; observe clear pattern on the *Z* axis; (**d**) scores provided by squats’ detector and post-processing; blue vertical rectangles represent ground-truth events and the gray rectangle represents single detection. The detection of repetition is counted if the score exceeds the detection threshold (upper dashed line). To count the next detection, the score must fall below the background threshold (lower dashed line) and again exceed the detection threshold. Any two consecutive detections must be separated by at least the refraction time (width of the gray rectangle). The detection event matches with the beginning of the gray rectangle.

**Figure 2 sensors-24-03891-f002:**
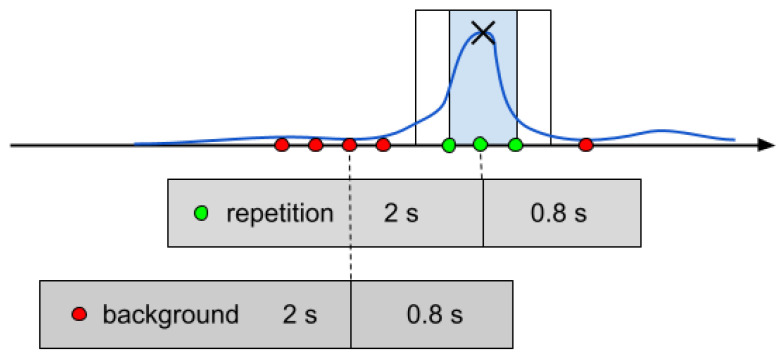
Diagram of the data window: cross denotes fiducial point of repetition; blue rectangle denotes annotation; a margin surrounds it (no example is taken from this margin); red and green dots denote points that provide labels for data windows (depicted as gray rectangles); we labeled the window as ‘repetition’ if a sample at the 2nd s falls into annotation (see dashed line); the upper rectangle is a ‘repetition’ example, and the lower one is ‘background’. The data window had 2 s of history and 0.8 s of look ahead. The construction of the data window leads to a detection latency of 0.8 s, which is acceptable, according to our experiments. We took 3 repetition examples from each annotation (at 0 s, 0.1 s, and 0.2 s) and took background examples from the inter-annotation space with 0.1 s steps. We left a 0.1 s margin around each annotation.

**Figure 3 sensors-24-03891-f003:**
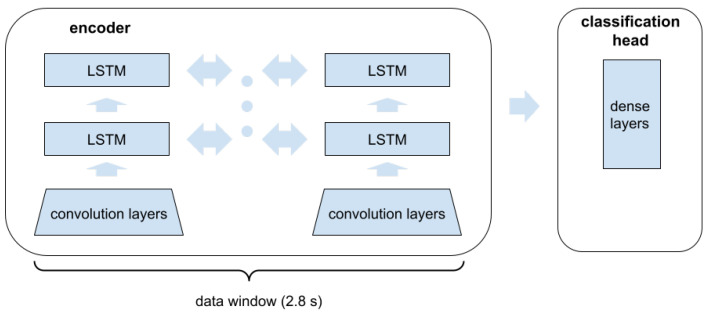
Block diagram of the neural scoring model for repetition detection. See Methods for description.

**Table 1 sensors-24-03891-t001:** Several human activity recognition systems based on IMU sensors.

Title	Task	Data Source	Activities	Method	Quality
Recognition and repetition counting for complex physical exercises with deep learning [2]	exercise recognition and repetition counting	signals recorded simultaneously from 2 smartwatches	10 complex full-body exercises typical in CrossFit (e.g., pull-ups, push-ups, burpees)	two separate models for exercise recognition and the start of repetition detection; deep CNN; overlapping 5 s data window; offline	recognition accuracy: 99.96%; repetition counting: ±1 repetitions in 91% of the tests
Recognition and repetition counting for local muscular endurance exercises in exercise-based rehabilitation: A comparative study using artificial intelligence models [1]	exercise recognition and repetition counting	single wrist-worn IMU sensor	10 endurance-based exercises (e.g., biceps curls, squats, lunges)	recognition task: multi-class classification with a deep CNN based on AlexNet architecture; repetition counting: counts compact segments of detection; offline	recognition F1-score: 97.18%; repetition counting: ±1 repetition error in 90% of the tests
Human activity recognition from accelerometer data using a wearable device [23]	activity recognition	single IMU sensor located on the chest	5 activities: regular walking, climbing stairs, talking with a person, staying standing, working at the computer	activity recognition: random forest; 20 features computed for 1 s data windows	activity recognition accuracy: 94%
myHealthAssistant: a phone-based body sensor network that captures the wearer’s exercises throughout the day [24]	exercise recognition and repetition counting	3 accelerometers (on the hand, arm, and leg)	13 exercises	exercise recognition: Bayesian classifier trained on the mean and variance on each accelerometer axis; repetition counting: peak counting on one of the accelerometer axes; offline	recognition accuracy: 92% (subject-specific model)
RecoFit: Using a wearable sensor to find, recognize, and count repetitive exercises [11]	segmenting exercise from intermittent non-exercise/rest periods; exercise recognition and repetition counting	accelerometer on the arm	26 exercises	segmentation and recognition tasks: linear support vector machines, features from 5 s data window; repetition counting performed offline with peak counting	segmentation precision and recall: >95%; exercise recognition accuracy: 96–99%; repetition counting ±1 repetition in 93% of the tests

**Table 2 sensors-24-03891-t002:** Quality of detectors with respect to the exercise and the training method.

Exercise (Training)	F1 (%)	MAPE (%)	FPR (Events/s)
abd. tenses (enc)	97 (1)	2 (2)	0.03 (0.01)
abd. tenses (e2e)	97 (1)	0 (2)	0.08 (0.02)
dw.-dog (enc)	58 (4)	72 (23)	0.10 (0.08)
dw.-dog (e2e)	64 (6)	67 (30)	0.16 (0.11)
lying hip rises (enc)	98 (1)	1 (1)	0.00 (0.01)
lying hip rises (e2e)	99 (1)	0 (1)	0.02 (0.01)
side lunges (enc)	98 (5)	4 (5)	0.00 (0.01)
side lunges (e2e)	88 (3)	13 (6)	0.02 (0.02)
sit-to-stands (enc)	92 (1)	8 (3)	0.02 (0.02)
sit-to-stands (e2e)	87 (2)	21 (7)	0.07 (0.01)
bends (enc)	86 (1)	21 (2)	0.03 (0.01)
bends (e2e)	68 (8)	41 (11)	0.13 (0.08)
broad jumps (enc)	99 (1)	1 (1)	0.02 (0.02)
broad jumps (e2e)	99 (1)	0 (1)	0.12 (0.03)
burpees (enc)	89 (2)	5 (2)	0.01 (0.01)
burpees (e2e)	87 (6)	2 (4)	0.23 (0.05)
crunches (enc)	92 (2)	5 (3)	0.04 (0.01)
crunches (e2e)	93 (1)	5 (1)	0.14 (0.02)
lunges (enc)	99 (3)	1 (4)	0.02 (0.02)
lunges (e2e)	99 (2)	1 (3)	0.06 (0.02)
push-ups (enc)	71 (6)	36 (8)	0.04 (0.04)
push-ups (e2e)	25 (4)	81 (4)	0.32 (0.16)
squats (enc)	88 (2)	7 (5)	0.04 (0.02)
squats (e2e)	77 (4)	15 (10)	0.11 (0.07)

## Data Availability

Data supporting reported results are available at https://www.aidlab.com/datasets. Data was described in Section 3.1.

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
