# Peer review of "Real-Time Sensor-Based Human Activity Recognition for eFitness and eHealth Platforms"

_sensors, 2024, doi:10.3390/s24123891_

Round 1

Reviewer 1 Report

Comments and Suggestions for Authors

The manuscript presents a method for sensor-based human activity recognition. In general terms the paper is well structured. However, it is looks like it was not edited. Across the text there are many instances of words with extra space inside the word, like " im portant " in Line 10 and so on. Also, subsection labels are the plain text. Also, for enumerated items (like in Lines 140-145) it is better to place one point in one line. Also, Fig. 1 has quite low resolution and unreadable.

There are lots of papers related to  studying the task of HAR. The manuscript presents a deep learning approach based on CNN+LSTM. This method is tested on one dataset, which was collected by the Authors.

The main problem of the paper is that there is no comparison with existing methods of HAR. Also, nothing is said about the efficiency of the presented method for some other datasets for HAR. Also, only one type of net architecture is used with some specific sizes of layers. It is not clear whether this architecture is optimal at least empirically?

These points make the manuscript to look like: one specific new approach for one specific new dataset. I guess that for Q1 journal this is not enough and the manuscript can be reconsidered after adding additional experiments.

Also, the link to the dataset doesn't work.

Comments on the Quality of English Language

No special comments

Author Response

"The manuscript presents a method for sensor-based human activity recognition. In general terms, the paper is well structured. However, it looks like it was not edited. Across the text there are many instances of words with extra space inside the word, like " im portant " in Line 10 and so on. Also, subsection labels are the plain text."

The paper is written in standard Latex using Sensors template, we do not see any formatting problems.

"Also, for enumerated items (like in Lines 140-145) it is better to place one point in one line. Also, Fig. 1 has quite low resolution and unreadable."

We fixed these issues.

"There are lots of papers related to  studying the task of HAR. The manuscript presents a deep learning approach based on CNN+LSTM. This method is tested on one dataset, which was collected by the Authors.

The main problem of the paper is that there is no comparison with existing methods of HAR. Also, nothing is said about the efficiency of the presented method for some other datasets for HAR. Also, only one type of net architecture is used with some specific sizes of layers. It is not clear whether this architecture is optimal at least empirically?"

The project aimed to develop a practical system including an exercise detector and data from the recorder placed on the chest. We have added a table summarizing a few studies and written about the advantages of our approach. 
We also provide information about data collection protocol, training, and evaluation. The data is now here:

https://www.aidlab.com/datasets

We have shown how particular design decisions, such as contrastive learning, help in building detectors.
We tested other neural network architectures and other classification models. However, they gave worse results, so we decided not to include them in the paper. We have extended comments on the other architectures and models.

"These points make the manuscript to look like: one specific new approach for one specific new dataset. I guess that for Q1 journal this is not enough and the manuscript can be reconsidered after adding additional experiments."

Development of models for a particular dataset collected by authors looks like standard practice in Sensors:
https://www.mdpi.com/1424-8220/20/17/4791
https://www.mdpi.com/1424-8220/19/3/714
We have not tested our models/learning method on another dataset because of the difference in location and the type of the recorder. We did not try to make a review paper on HAR. Our primary goal was to create a practical system for mobile devices using signals from a particular recorder. 
The knowledge about the performance of our model on other datasets would not help in the project.
We also have not reported the results of the other models on our data sets since we are interested in online/real-time models on mobile devices.

"Also, the link to the dataset doesn't work."
Sorry for that; the correct link is now here:

https://www.aidlab.com/datasets

Reviewer 2 Report

Comments and Suggestions for Authors

Overall this paper provides a very useful reporting of the noted research performed and its unique methods involving a single IMU, chest mounted, in detecting various Human Activity Recognition.  Minor changes in the background to improve ease of readability would be useful in a table providing the previous actigraphy studies (as opposed to narrative paragraph form). 

Line 71 of the manuscript through line 86 of the background appear to be content better suited for inclusion in the Methods section, as this addresses the current study, consider moving. 

The discussion section, could be expanded to provide more content.  Specifically, lines 344 to 350 could further relate previous works to this project.

Comments on the Quality of English Language

A few minor typographical errors noted in spelling and the use of third person, as opposed to first person throughout the manuscript, would be more appropriate. 

Author Response

"Overall this paper provides a very useful reporting of the noted research performed and its unique methods involving a single IMU, chest mounted, in detecting various Human Activity Recognition. Minor changes in the background to improve ease of readability would be useful in a table providing the previous actigraphy studies (as opposed to narrative paragraph form)."

We decide to stay with the narrative paragraph form.

"Line 71 of the manuscript through line 86 of the background appear to be content better suited for inclusion in the Methods section, as this addresses the current study, consider moving. "Thank you, we have followed this suggestion.

"The discussion section could be expanded to provide more content. Specifically, lines 344 to 350 could further relate previous works to this project."

We have expanded the discussion considerably.
A correct link to our data sets, and a new table with selected previous HAR studies have been added.

"A few minor typographical errors noted in spelling and the use of third person, as opposed to first person throughout the manuscript, would be more appropriate. "

Indeed, in the past, third-person and passive forms were standard. We reviewed the manuscript but decided to stay with the first person as these days it is a common practice in Sensor.

Round 2

Reviewer 1 Report

Comments and Suggestions for Authors

The issues from my first report were mostly addressed by appropriate changes in the manuscript and by comments in the authors' responses. The paper now looks more informative and clear. I do not have any further comments. I guess that the paper can now be accepted.